# Movement side effects of antipsychotic drugs in adults with and without intellectual disability: UK population-based cohort study

Rory Sheehan,[1] Laura Horsfall,[2] André Strydom,[1] David Osborn,[1] Kate Walters,[2] Angela Hassiotis[1]

► Prepublication history and additional material is available. To view please visit the journal (http://dx.doi.org/ 10.1136/bmjopen-2017-017406).

RS and LH contributed equally.

## ABSTRACT

**Objectives** To measure the incidence of movement side effects of antipsychotic drugs in adults with intellectual disability and compare rates with adults without intellectual disability.

**Design** Cohort study using data from The Health Improvement Network.

**Setting** UK primary care.

**Participants** Adults with intellectual disability prescribed antipsychotic drugs matched to a control group of adults without intellectual disability prescribed antipsychotic drugs.

**Outcome measures** New records of movement side effect including acute dystonias, akathisia, parkinsonism, tardive dyskinaesia and neuroleptic malignant syndrome.

**Results** 9013 adults with intellectual disability and a control cohort of 34 242 adults without intellectual disability together contributed 148 709 person-years data. The overall incidence of recorded movement side effects was 275 per 10 000 person-years (95% CI 256 to 296) in the intellectual disability group and 248 per 10 000 person-years (95% CI 237 to 260) in the control group. The incidence of any recorded movement side effect was significantly greater in people with intellectual disability compared with those without (incidence rate ratio 1.30, 95% CI 1.18 to 1.42, p<0.001, after adjustment for potential confounders), with parkinsonism and akathisia showing the greatest difference between the groups. Neuroleptic malignant syndrome, although occurring infrequently, was three times more common in people with intellectual disability-prescribed antipsychotic drugs (incidence rate ratio 3.03, 95% CI 1.26 to 7.30, p=0.013). Differences in rates of movement side effects between the groups were not due to differences in the proportions prescribed first and second-generation antipsychotic drugs.

**Conclusions** This study provides evidence to substantiate the long-held assumption that people with intellectual disability are more susceptible to movement side effects of antipsychotic drugs. Assessment for movement side effects should be integral to antipsychotic drug monitoring in people with intellectual disability. Regular medication review is essential to ensure optimal prescribing in this group.

[1]Division of Psychiatry, University College London, London, UK
[2]Research Department of Primary Care and Population Health, University College London, London, UK

**Correspondence to**
Dr Rory Sheehan;
r.sheehan@ucl.ac.uk

## Strengths and limitations of this study

► This study includes a very large number of people with and without intellectual disability who have been prescribed antipsychotic drugs.
► The Health Improvement Network is a UK primary care database that contains accurate recording of demographic and clinical information and drug prescribing.
► This is the first study to directly compare the rates of movement side effects of antipsychotic drugs between people with intellectual disability and those without, and offers new insights into the risk:benefit ratio of antipsychotic drug prescribing to people with intellectual disability.
► Recording of movement side effects of antipsychotic drugs in primary care has not been validated.
► Antipsychotic drugs prescribed outside primary care and movement side effects identified in other settings may not have been recorded by our method.

## INTRODUCTION

Movement (extrapyramidal) side effects, including acute dystonias, akathisia, parkinsonism and tardive dyskinaesia are a well-recognised complication of antipsychotic drugs which are thought to occur secondary to antagonism of dopamine D2 receptors in the striatum and mesocortex.[1] Movement side effects can be distressing, disabling and difficult to treat and their presence is associated with poor medication compliance, stigma and reduced quality of life.[2]

Intellectual disability (ID) is a lifelong condition characterised by global deficits in cognitive and adaptive functioning. People with ID experience relatively high rates of mental illness[3] and many are prescribed antipsychotic drugs in the UK and worldwide. There has been renewed focus on the appropriateness of antipsychotic drug prescribing in people with ID following recent evidence that antipsychotic drugs are often used in

BMJ

the absence of an underlying diagnosed mental illness,[4] in many cases in an attempt to manage challenging behaviour, despite a lack of evidence that they are effective in this context.[5] There has been relatively little formal investigation of antipsychotic drug side effects in people with ID and most of our knowledge is extrapolated from studies conducted in people of average intelligence. People with ID are often considered to be at greater risk of antipsychotic drug-induced movement side effects than people without ID[6] but no studies directly compare rates between the two groups. Furthermore, knowledge of a specific mechanism that might underpin any association between ID and movement side effects extends only to a vague theory that organic brain dysfunction makes centrally mediated side effects of psychotropic drugs more likely.[7]

We undertook a cohort study using a large nationally representative database to compare the incidence of recorded movement side effects in adults with and without ID who are prescribed antipsychotic drugs.

## METHODS
### Data source
Data were obtained from The Health Improvement Network (THIN), a large UK primary care database that contains the electronic health records of more than 3.7 million active patients in over 550 general practices (http://www.epic-uk.org/our-data/our-data.shtml). The patients included in the database are representative of the UK population in age, sex and morbidity and mortality.[8 9] The vast majority of people with intellectual disability in the UK live in the community and are registered with a general practitioner (GP; primary care physician) who provides routine and ongoing care and who acts as gatekeeper for hospital-based specialists, including psychiatrists. The THIN database contains clinical records added by GPs using a clinical dictionary of Read codes. Read codes are standardised clinical terms that can be used as shorthand for clinicians to record certain patient characteristics (such as occupation and living circumstances) and the content of a consultation.[10] Individual Read codes exist to cover the variety of signs, symptoms and diagnoses that an individual may have, as well as test results and surgical or therapeutic treatments. Recording of illness in primary care records has been shown to be accurate and all prescribed medication must be issued through the electronic system. National Health Service drug budgets flow through primary care and GPs issue most prescriptions directly, including those for psychotropic drugs. The primary care record therefore is a suitable means of conducting pharmacoepidemiological research.

THIN data are pseudonymised at source and made available to researchers who have purchased a license. THIN has overall ethical approval to collate data and this study received approval from the THIN Scientific Review Committee (reference 14–071).

### Study cohort
For this study, all adults with recorded ID and a history of oral antipsychotic drug prescription were extracted using a previously defined and tested list of diagnostic Read codes (including codes for ID and conditions associated with ID) and antipsychotic drug codes,[4] based on chapters of the British National Formulary. General practitioners are incentivised by the Quality Outcomes Framework to keep a register of people with ID which improves recording in the database. The study period was 1 January 1999 to 31 December 2014. Entry to the cohort was set as the date of the first antipsychotic drug prescription issued after the latest of; registration with the GP practice contributing data; the patient's 18th birthday; the start of the study period; the date the practice achieved compliance with standard measures of data quality.[11 12] People contributed person-years (PYs) of data from entry to cohort exit. Exit from the cohort was defined as the first of: the final antipsychotic drug prescription plus the length of the prescription; deregistration with the GP practice contributing data; the end of the study period or death.

A comparison cohort of people prescribed antipsychotic drugs but without ID was extracted using stratified sampling within each GP practice with frequency matching to ensure similar population-level characteristics across the two cohorts in terms of age, gender and year of antipsychotic prescription. Up to six people without ID were selected for every person with ID and the same criteria were used to define cohort entry and exit.

A Read code list for movement side effects was developed using previously described methodology and applied to the cohort to determine incidence of movement side effects[13] (see online supplementary data). Movement side effects were categorised as acute dystonias, akathisia, parkinsonism or tardive dyskinaesia in accordance with orthodox classification. A separate category was established for neuroleptic malignant syndrome, being a very specific adverse effect and a further category included for broad codes which could not be sub-categorised. Prescriptions for selected antimuscarinic drugs (procyclidine, orphenadrine, trihexyphenidyl) were used as proxy indicators of movement side effects in those prescribed antipsychotic drugs. People were defined as having a history of movement disorder if a relevant Read code was applied (or antimuscarinic drug prescribed) prior to cohort entry or within 6 months of registration with the practice, as this has been shown to improve the validity of incidence calculations.[14]

### Covariates
Sociodemographic covariates included age, sex, calendar year and the Townsend Deprivation Score (a composite score in fifths based on postcode and census recording of local unemployment, car ownership, home ownership and overcrowding).[15] Other covariates included history of antipsychotic use at cohort entry, antipsychotic average daily dose and days on treatment between the start and

end of follow-up. Average daily dose was measured as chlorpromazine equivalents to account for polypharmacy and those who switched drugs during follow-up. Where we were unable to extract daily dose data (for example, in the minority of cases where the duration of a prescription was not recorded) the prescribed dose for the previous or subsequent prescription for the same drug and formulation was used. It was not possible to estimate the daily dose for 5% of prescriptions, and these were excluded from the study. Prescriptions for drug doses above three times the upper licensed limit were excluded as probable coding errors (<1%).

## Statistical analysis

Multivariable mixed Poisson regression was used to calculate incident rates of movement disorders during exposure to antipsychotic drugs in people with and without ID by calendar year, adjusted for any temporal changes in age and sex. Incidence rate was defined as the number of new events of interest/the duration that the cohort was at risk. First we were interested in the incidence of new cases of any movement disorder. Participants exited the cohort when they were first diagnosed with any movement disorder as they were no longer considered at risk of a new diagnosis after this date. For calculating the incidence of subtypes of movement side effect, participants exited the cohort after they were diagnosed with the type of movement side effect of interest as they were no longer considered at risk of that type of movement side effect after that date. They remained in the cohort for the purposes of being diagnosed with other types of movement side effect as an individual participant may develop more than one type of movement side effect of antipsychotic drug.

General practice was included as a random effect to account for any data clustering. Calendar year was initially modelled as a continuous variable and we then used the likelihood ratio test to compare this with a model in which year was entered as a categorical variable to examine the possibility of non-linear time trends.

Multivariable mixed Poisson regression was also used to compare the rates of movement disorder during exposure to antipsychotic drugs in people with and without ID, adjusted for covariates. We conducted a sensitivity analysis where we restricted the analysis to time periods when people were exclusively prescribed first or second-generation antipsychotic drugs, and further when we restricted the analysis to times when only risperidone was prescribed. All analyses were repeated after excluding people with a diagnosis of idiopathic Parkinson's disease.

We considered a p value of 0.05 to be statistically significant (two-tailed) and used Stata V.13 for all analyses (StataCorp).

## RESULTS

In total, 9039 people with ID met inclusion criteria and were matched to 34 242 people without ID, and together contributed 148 709 PYs data. The two cohorts were similar in terms of age, sex, level of social deprivation and history of movement disorder at cohort entry (table 1). The prevalence of movement disorder at baseline was slightly higher for people with ID and a history of antipsychotic use (31%) compared with those without ID and a history of antipsychotic drug use (24%) but the proportions of those with a history of movement disorder without antipsychotic drug use were equal (6%) at cohort entry. Average daily dose of antipsychotic was similar between the two groups but those with ID had longer time periods between their first and last antipsychotic prescription and more days on treatment between those dates. Table 2 shows the distribution of antipsychotic drugs prescribed to the study cohort. Risperidone was the most common drug prescribed in the ID cohort (28.5% prescriptions to people with ID, 14.7% prescriptions in the non-ID cohort); other drugs were prescribed in roughly equal proportions between the two groups.

The overall incidence of recorded movement side effects was 275 per 10 000 PYs (95% CI 256 to 296) in the ID group and 248 per 10 000 PYs (95% CI 237 to 260) in the non-ID group (table 3). Parkinsonism was the most commonly recorded movement side effect in both groups. After adjustment, the incidence rate of any movement disorder was 30% higher in people with ID compared with those without ID (incidence rate ratio 1.30, 95% CI 1.18 to 1.42, p<0.001). Similar differences in movement side effect recording were noted when defined by diagnostic Read codes or by proxy, using prescription for antimuscarinic drugs. The incidence rates of akathisia, parkinsonism and neuroleptic malignant syndrome were significantly higher in those with ID compared with those without ID.

Analysis restricted to periods when people were exclusively prescribed first or second-generation antipsychotic drugs did not change the results; movement side effects were still significantly more likely to be recorded in people with ID compared with those without (table 4). Excluding people with Parkinson's disease from the analysis (n=451) had no meaningful effect on any of the results.

Time trend analysis showed that the incidence of recording of movement side effects in those prescribed antipsychotic drugs fell significantly over the course of the study period in both groups (figure 1); each calendar year was associated with a 5% decline in the recording of movement side effects in people with ID (95% CI 2% to 8%, p<0.0001) and a 7% decline in people without ID (95% CI 5% to 9%, p<0.001), after accounting for changes in cohort age and sex. Prescriptions for antimuscarinic drugs declined by 3% per year in people with ID (95% CI 1% to 5%, p=0.002) and 5% per year in people without ID (95% CI 4% to 7%, p<0.001). Average daily antipsychotic dose, measured in chlorpromazine equivalents, remained broadly constant during the study period (data not shown).

**Table 1** Cohort characteristics

|  | ID cohort | Non-ID cohort |
|---|---|---|
| Total number (%) | 9039 (21) | 34 242 (79) |
| Male, n (%) | 5279 (58) | 18 825 (55) |
| Average age, years (SD) | 42 (16) | 44 (16) |
| Townsend score, n (%) |  |  |
| 1 | 1403 (16) | 4860 (14) |
| 2 | 1752 (19) | 5119 (15) |
| 3 | 1948 (22) | 6481 (19) |
| 4 | 1942 (22) | 8189 (24) |
| 5 | 1563 (17) | 7911 (23) |
| Missing | 417 (5) | 1619 (5) |
| History of antipsychotic use at cohort entry, n (%) | 6684 (74) | 16 227 (47) |
| History of movement disorder at cohort entry, n (%) | 2192 (24) | 4946 (14) |
| History of movement disorder without antipsychotic use at cohort entry, n (%) | 136/2355 (6) | 1038/18 015 (6) |
| History of movement disorder and antipsychotic use at cohort entry, n (%) | 2056/6684 (31) | 3908/16 227 (24) |
| Total person-years between first and last antipsychotic prescription | 44 696 | 104 014 |
| Median years between first and last prescription (IQR) | 3.5 (1.2 to 7.9) | 1.3 (0.19 to 4.4) |
| Median years on treatment between first and last prescription (IQR) | 2.6 (0.76 to 6.3) | 0.67 (0.15 to 2.5) |
| Average daily dose, CLZE (SD) | 135 (156) | 139 (146) |

CLZE, chlorpromazine equivalents; ID, intellectual disability.

## DISCUSSION

People with ID have long been considered at greater risk of adverse side effects of antipsychotic drugs. However to date, very little evidence has been presented to substantiate this belief. Our data suggest that people with ID are more likely to experience movement side effects of antipsychotic drugs; this finding is robust and persists when movement side effects are defined by diagnostic Read code alone and when they are measured using prescription of antimuscarinic drugs as proxy.

Most people in our cohort received a second-generation antipsychotic drug. The types of drugs that were prescribed in our study were broadly similar between the group with ID and the group without, and were consistent with other recent data examining antipsychotic prescribing in community-dwelling adults living in the UK.[16] The exception was risperidone, which was prescribed more frequently to people with ID; this accords with other studies[17] and is likely to reflect an attempt by clinicians to prescribe the antipsychotic with greatest (although still limited) evidence of benefit for challenging behaviour in people with ID.[5 18] The difference in incidence of movement side effects between people with and without ID remained when first and second-generation agents were considered separately, and when risperidone was considered alone, suggesting that headline differences between the groups were not due to different prescribing practices.

Although their improved movement side effect profile has been considered a major advantage of second-generation antipsychotic drugs, more recent evaluation of the evidence suggests that the initial enthusiasm for second-generation agents was misplaced and largely based on studies that made unequal comparisons between second-generation and high-potency first-generation drugs.[19 20] In this study, we did not set out to compare movement side effects between first and second-generation agents but we observed that the prescription of second-generation agents was associated with a slightly lower incidence of recorded movement side effects in people with and without ID; clearly further work is needed to provide definitive data on this contentious aspect of antipsychotic drug side effects.

There is a lack of work with which we can directly compare our results; previous studies that investigate movement disorders in people with ID who take antipsychotic drugs have used differing methods to define and ascertain movement disorder, selected particular populations (often convenience sampling those residing in institutions) and tend to report point prevalence figures. None have directly compared rates of movement side effect in people with ID to controls without ID. Nevertheless, it is clear that antipsychotic drug-induced movement side effects are reasonably common in people with ID. In once recent study of hospitalised patients with borderline-mild ID and challenging behaviour, almost

**Table 2** Frequency of antipsychotic drugs prescribed in the ID and non-ID cohorts

| Antipsychotic drug | ID cohort | | | | Non-ID cohort | | | |
|---|---|---|---|---|---|---|---|---|
| | Number | Percentage | Average daily dose, mg | Median treatment duration, years | Number | Percentage | Average daily dose, mg | Median treatment duration, years |
| Risperidone | 4013 | 28.5 | 1.9 | 1.81 | 7426 | 14.7 | 2.2 | 0.47 |
| Olanzapine | 2086 | 14.8 | 8.2 | 1.62 | 10246 | 20.3 | 8.5 | 0.67 |
| Chlorpromazine | 1770 | 12.6 | 78.5 | 1.12 | 5274 | 10.5 | 64.8 | 0.20 |
| Quetiapine | 1295 | 9.2 | 154.1 | 0.93 | 7693 | 15.2 | 152.6 | 0.72 |
| Haloperidol | 1231 | 8.7 | 4.9 | 0.68 | 3755 | 7.5 | 3.2 | 0.14 |
| Thioridazine | 838 | 6.0 | 77.9 | 0.82 | 2672 | 5.3 | 49.8 | 0.20 |
| Aripiprazole | 661 | 4.7 | 10.5 | 0.74 | 2638 | 5.2 | 11.8 | 0.54 |
| Trifluoperazine | 456 | 3.2 | 6.5 | 1.06 | 3403 | 6.8 | 4.5 | 0.19 |
| Zuclopenthixol | 429 | 3.1 | 18.7 | 2.17 | 340 | 0.7 | 20.0 | 0.45 |
| Amisulpride | 327 | 2.3 | 295.0 | 0.94 | 1687 | 3.4 | 290.3 | 0.73 |
| Promazine | 276 | 2.0 | 58.4 | 0.33 | 1688 | 3.4 | 58.6 | 0.16 |
| Sulpiride | 276 | 2.0 | 437.8 | 1.97 | 1199 | 2.4 | 435.6 | 0.79 |
| Other* | 980 | 7.0 | – | – | 5230 | 10.4 | – | – |

*Other antipsychotic drugs prescribed to <1% of ID cohort each.
ID, intellectual disability.

half were found to have a movement disorder, and the presence of movement disorder was more likely in those prescribed antipsychotic medication.[21] The most common type was parkinsonism, as in our study. De Kuijper and colleagues in the Netherlands report that just over half of their sample with ID who had been taking antipsychotic drugs for more than a year had evidence of movement side effects.[22]

It is interesting that in our study the rates of recording of tardive dyskinaesia were relatively low and it was not more frequently recorded in people with ID. Previous work has shown tardive dyskinaesia to be common; one study of institutionalised adults with ID taking long-term antipsychotic drugs found a prevalence rate of tardive dyskinaesia of 45%.[23] Spontaneous dyskinaesias are also common in people with ID[24] and it is possible that drug-induced tardive dyskinaesia may be misinterpreted as part of the underlying ID and under-recorded, an example of 'diagnostic overshadowing'. Conversely, it is also possible that background dyskinaesia related to ID might be misinterpreted as being the result of antipsychotic drugs. Several assessment scales are available for measuring movement side effects of antipsychotic drugs and may be utilised in monitoring, although there are obvious challenges in assessment of subjective symptoms (such as akathisia) people with ID who may have limited understanding and verbal communication ability.[25–27]

Neuroleptic malignant syndrome is a rare idiosyncratic complication of antipsychotic therapy consisting of fever, muscle rigidity, autonomic dysfunction and alterations in cognitive state. We found a significantly increased incidence of neuroleptic malignant syndrome among people with ID, although the low number of recorded events in the database means our results need to be interpreted with caution. An association between neuroleptic malignant syndrome and ID has been demonstrated previously[28 29] and this, combined with the seriousness of the condition (particularly in people with ID,[29 30] warrants further attention.

We observed a decline in the recording of movement side effects in both groups over the past 15 years. It might be that clinicians have focused their attention on measuring and managing metabolic complications; the wide scale switch from first to second-generation antipsychotic drugs[31] has partially contributed to an actual decrease in the rate of movement side effect; the clinical expectation of reduced movement side effects with newer drugs has reduced vigilance and recognition of these side effects.

### Strengths and limitations

This is the first study to directly compare the rate of antipsychotic drug-induced movement side effects between people with and without ID. Our findings are strengthened by the large numbers of people included and the sample being drawn from a representative community population. UK general practices are incentivised to maintain an accurate list of people with ID, and as

**Table 3** Incidence rates and adjusted incident rate ratios for movement side effects in people with and without ID prescribed antipsychotic drugs

| Variable | ID cohort | | | Non-ID cohort | | | Comparison | |
|---|---|---|---|---|---|---|---|---|
| | Events during follow-up (n) | Person-years (×10 000) (n) | Incidence per 10 000 person-years (95% CI) | Events during follow-up (n) | Person-years (×10 000) (n) | Incidence per 10 000 person-years (95% CI) | Incidence rate ratio* (95% CI) | p Value |
| Any movement disorder (defined by Read code or antimuscarinic prescription) | 743 | 2.7 | 275 (256 to 296) | 1750 | 7.0 | 248 (237 to 260) | 1.30 (1.18 to 1.42) | <0.001 |
| Any movement disorder (defined by Read code) | 446 | 4.4 | 111 (101 to 122) | 952 | 9.4 | 101 (95 to 108) | 1.30 (1.16 to 1.47) | <0.001 |
| Any movement disorder (defined by antimuscarinic prescription) | 564 | 2.9 | 196 (180 to 212) | 1299 | 7.6 | 172 (163 to 181) | 1.29 (1.16 to 1.44) | <0.001 |
| Acute dystonia | 60 | 4.4 | 14 (11 to 18) | 161 | 10.2 | 16 (14 to 19) | 1.00 (0.73 to 1.37) | 0.99 |
| Akathisia | 80 | 4.5 | 18 (15 to 23) | 112 | 10.3 | 11 (9 to 13) | 2.29 (1.69 to 3.12) | <0.001 |
| Parkinsonism | 270 | 4.4 | 64 (57 to 72) | 592 | 9.9 | 60 (55 to 65) | 1.20 (1.03 to 1.39) | 0.02 |
| Tardive dyskinaesia | 61 | 4.0 | 14 (11 to 18) | 123 | 10.3 | 12 (10 to 14) | 1.27 (0.91 to 1.75) | 0.16 |
| Neuroleptic malignant syndrome | 11 | 4.4 | 3 (1 to 5) | 12 | 10.4 | 1 (1 to 2) | 3.03 (1.26 to 7.30) | 0.013 |
| Other movement disorder | 43 | 4.2 | 10 (7 to 13) | 94 | 10.3 | 9 (7 to 11) | 1.26 (0.86 to 1.85) | 0.23 |

*Adjusted for sex, social deprivation score, time period, history of antipsychotic drug use, average daily dose, days on treatment.
ID, intellectual disability.

**Table 4** Sensitivity analysis with incidence rates and adjusted incident rate ratios for movement side effects in people with and without ID prescribed first and second-generation antipsychotic drugs

| Antipsychotic class | ID cohort | | | Non-ID cohort | | | Comparison | |
|---|---|---|---|---|---|---|---|---|
| | Events during follow-up (n) | Person-years (×10 000) (n) | Incidence per 10 000 person-years (95% CI) (n) | No of events during follow-up | Person-years (×10 000) (n) | Incidence per 10 000 person-years (95% CI) | Incidence rate ratio* (95% CI) | p Value |
| First generation† | 247 | 0.8 | 320 (283 to 362) | 569 | 1.9 | 293 (270 to 318) | 1.36 (1.16 to 1.60) | <0.001 |
| Second generation† | 378 | 1.6 | 241 (218 to 267) | 948 | 4.3 | 219 (206 to 233) | 1.43 (1.26 to 1.62) | <0.001 |
| Risperidone† | 124 | 0.6 | 196 (164 to 233) | 96 | 0.5 | 182 (149 to 223) | 1.55 (1.15 to 2.08) | 0.004 |

*Adjusted for sex, social deprivation score, time period, history of antipsychotic drug use, average daily dose, days on treatment.
†Restricted to periods when people were exclusively prescribed first or second-generation antipsychotic drugs or risperidone.
ID, intellectual disability.

prescriptions are also accurately recorded in THIN, our results are generalisable across settings to all people with intellectual disability who take antipsychotic drugs. We excluded depot antipsychotic preparations (and the small number of prescriptions that may have been issued in secondary care) and therefore might have slightly underestimated exposure to antipsychotic drugs.

A limitation of our work that is common across observational studies that the use routinely collected health data is the lack of direct validation of diagnoses. The Read code list for movement side effects was devised using a comprehensive methodology and with input from practising primary and specialist secondary care physicians but not tested against 'gold standard' methods for identifying movement side effects. We assume that relevant Read codes added to patient records during exposure to antipsychotic drugs represent adverse side effects; this may not always be the case and symptoms of movement disorder may arise independently or in response to other prescribed medications (such as antidepressants and antiepileptic drugs) that we did not measure. Our method measures only recorded side effects, that is, people must consult their GP for the side effect to be noted formally. Even when seen by a clinician, there is evidence that movement side effects might be missed.[32] It is possible, therefore, that our results underestimate the true rate of movement side effect of antipsychotic drugs. How this might bias the comparison between ID and non-ID groups is not clear. People with ID have lower health literacy, lack knowledge of psychotropic drug side effects[33] and may encounter barriers to accessing primary care[34] and hence be less likely to present to primary care when experiencing treatment side effects. Conversely, people with ID may be monitored more closely by carers or by proactive GPs who recognise the higher health need in this group, for example, by offering an annual health check. Some cases of movement side effects may have not been recorded in the primary care database if people who are in contact with specialist services contact their psychiatrist directly rather than visiting their GP. Neuroleptic malignant syndrome may be underestimated either because milder forms are missed or because it is more likely to be treated in the acute hospital.

Further work will be needed to elucidate the potential pathophysiological mechanism underlying the observed association between ID and movement side effects of antipsychotic drugs.

## CONCLUSIONS

Movement side effects are only one aspect of a number of antipsychotic drug adverse effects. They can impact medication compliance, quality of life and compound the stigma of mental illness and/or intellectual disability.[35 36] They can be difficult to recognise, to treat and, in the case of the tardive syndromes, can persist or even worsen on withdrawal of the offending drug. People with ID appear more susceptible to movement side effects of antipsychotic

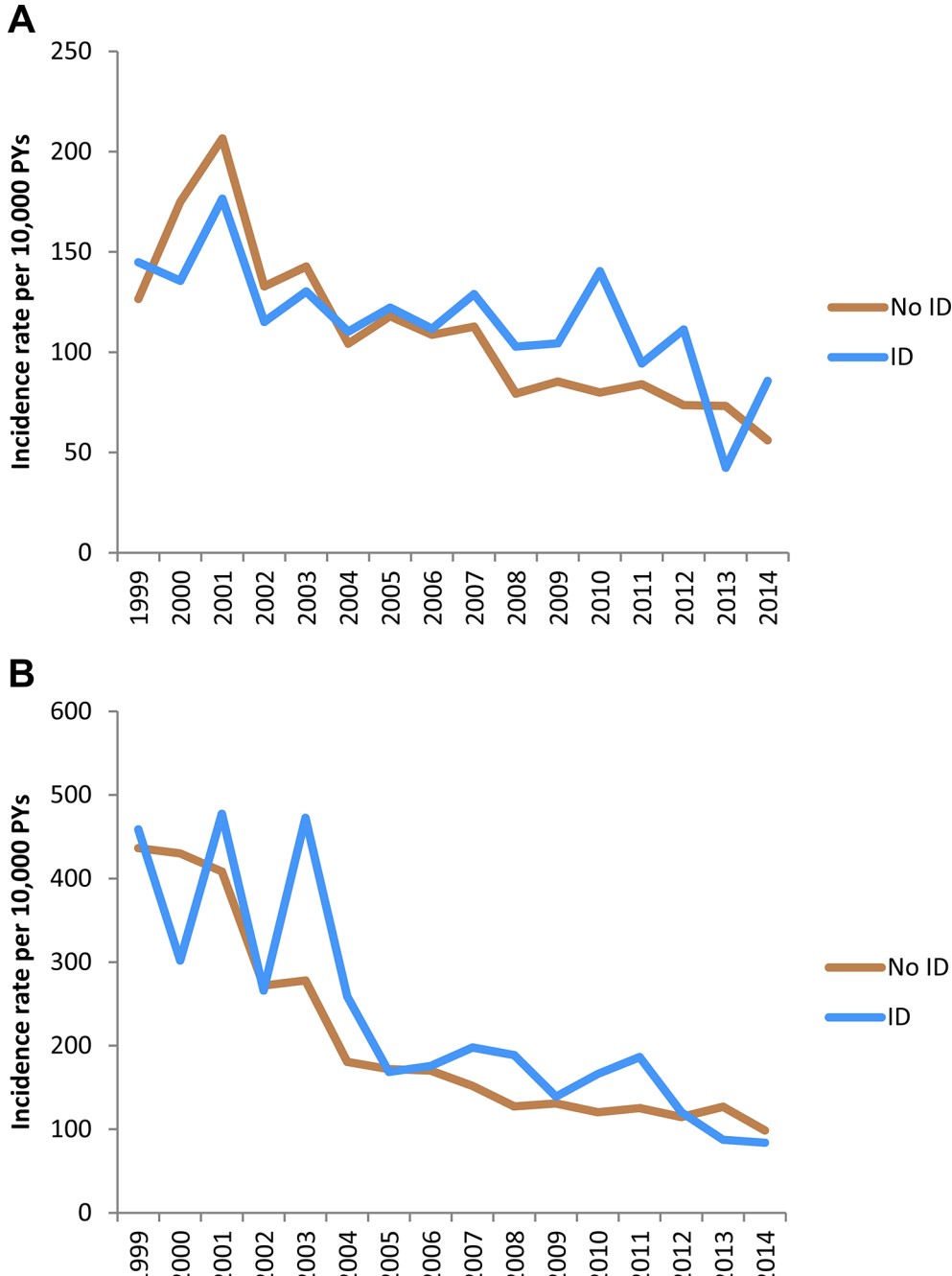

**Figure 1** Time trends in crude incidence rates of (A) movement side effect defined by Read code and (B) antimuscarinic drug prescription in people with and without ID prescribed antipsychotic drugs. ID, intellectual disability.

drugs than people without ID, and this should be considered when treatment decisions are made, especially given the relatively high rates of other comorbidities in this population. There is evidence that movement side effects of antipsychotic drugs are poorly assessed in people with ID who are under the care of secondary care services[37]; this situation must change if medication is to be used in the safest and most effective way possible.

Our data support a modest potential benefit of second-generation antipsychotic drugs in reducing the incidence of movement side effects, but more work is needed to confirm this finding, and it must be balanced against the increased propensity of second-generation agents to cause metabolic side effects.

There has been much recent public and professional interest in the prescription of antipsychotic drugs to people with intellectual disability and UK national policy supports attempts to reduce the prescribing of antipsychotic drugs for challenging behaviour (https://www.england.nhs.uk/wp-content/uploads/2016/06/stopping-over-medication.pdf). We recently showed that reduction or discontinuation of antipsychotic drugs in people with ID and challenging behaviour (but without severe mental illness) risks harm as well as providing

potential benefits and advocate individual treatment decisions in this group.[38] The current work informs the risk–benefit analysis undertaken as part of antipsychotic drug prescribing in people with ID and reinforces the need for regular and effective medication review, which must include assessment of possible movement side effects.

**Contributors** RS, LH, AS, DO, KW and AH developed the idea and method for the study, interpreted the results and wrote the manuscript. RS is the guarantor. LH performed the data extraction and analysis.

**Funding** This work was supported by The Baily Thomas Charitable Fund grant number 518669. The funder had no role in study design, data collection and analysis, decision to publish or preparation of the manuscript.

**Competing interests** None declared.

**Provenance and peer review** Not commissioned; externally peer reviewed.

**Data sharing statement** Copies of Read code lists used in this study are available from the authors, on request.

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
