## [Reviewer comments · BMJ Open]

ARTICLE DETAILS

TITLE (PROVISIONAL)	Movement side-effects of antipsychotic drugs in adults with and without intellectual disability: UK population-based cohort study
AUTHORS	Sheehan, Rory; Horsfall, Laura; Strydom, Andre; Osborn, David; Walters, Kate; HASSIOTIS, ANGELA

VERSION 1 - REVIEW

REVIEWER	Gerda de Kuijper, MD PhD Centre for intellectual disability and mental health/Mental health care Drenthe (CVBP/GGZ Drenthe) The Netherlands
REVIEW RETURNED	28-Apr-2017

GENERAL COMMENTS	This study contributes importantly to the knowledge concerning the occurrence of movement side-effects caused by antipsychotic drug use in people with ID, in particular to the knowledge regarding the assumed greater sensitivity of people with ID compared to people without ID. Data source is valid and by using stratified sampling with frequency matching the sample will be representative for people with and without ID who use antipsychotics. The results are presented clearly and are summarized in the tables and figure adequately. I have some questions and remarks. - In the Method section the dictionary of Read codes of the THIN database is mentioned. Since clinicians and researchers abroad may not be informed by the meaning of Read codes the authors are requested to explain this term.- Method section Page 8. line 11-14: People also excited at the date they developed a movement side-effect as they were no longer considered at risk after this time. Did the autors consider the possibility of the development of more/other movement side-effects after removal of the participant? How were incidence rates defined? The occurrence of one or more movement side-effects? Page 8, line 39-42. In the Netherlands the anticholinergic agent biperiden is widely used to decrease symptoms of drug-induced Parkinsonism. Do UK physicians also prescribe this agent in this indication? 6. Are the outcomes clearly defined? - In the Introduction section and in the Discussion section the word "Parkinsonism "is used to indicate the antipsychotic-induced Parkinson-disorder-like symptoms, while in the Method section the word "Pseudo-parkinsonism "is used. I would like to suggest to use
--

	the broadly accepted word "Parkinsonism" for antipsychotic-induced movement side-effects instead of the word "Pseudo-parkinsonism". Furthermore, I would like to receive the Read code list for movement side-effects. Discussion section : Page 13 , line 11-23 The rates of recording of tardive dyskinesia were relatively low.... The authors do not distinguish with regard to the level of severity of ID in the study sample. Studies have shown that more severe ID is associated with higher frequency and more severe symptoms of tardive dyskinesia. The mean severity of ID in this community dwelling study sample may be lower compared to other study samples of institutionalized participants. line 18-20...drug-induced tardive dyskinesia may be interpreted as part of the underlying ID and underrecorded, ... However, it may also be possible that the dyskinesia caused by the underlying ID is interpreted as drug-induced movement side-effect. 12. Are the study limitations discussed adequately? Page 14 line 37 In the manuscript is written that people with ID who use psychotropic drugs must consult their General Practitioner in case they suffer from drug-induced side-effects and that they may be less likely to present with side-effects compared to not ID people. Furthermore, the authors write that side-effects may have been missed if people with ID contact their psychiatrist directly. I wonder whether people with ID who use antipsychotics are being monitored for side-effects on a regular base in UK general practice and UK specialist practice. If not, the authors may provide some data on the number of people with ID who use antipsychotic drugs and consulted their physician for side-effects compared to those who use antipsychotics and did not consult their physician for side-effects.
--	---

REVIEWER	Tamara Pringsheim University of Calgary, Canada
REVIEW RETURNED	16-May-2017

GENERAL COMMENTS	This is a very clearly written paper about the incidence of extrapyramidal symptoms in individuals taking antipsychotic medications using the THIN database, with the focus on comparing rates in individuals with intellectual disability versus those without intellectual disability. This paper draws attention to the common, but now largely ignored issue of movement disorders in individuals exposed to antipsychotic medications. This paper demonstrates that not only are movement disorders more common in individuals with intellectual disability, but occur at a high frequency in all antipsychotic users, with 24% of the non-ID cohort having a history of movement disorder and
---

	antipsychotic use at cohort entry. They have also shown that the incidence of recording of movement side effects in those prescribed antipsychotics fell over the course of the 15 year study period in both groups. I believe this is a generational effect, with an entire cohort of psychiatrists and physicians trained in the second generation antipsychotic era who have poor recognition of movement disorders. I have no major concerns. I think this is a very important manuscript that deserves publication.
--	--

VERSION 1 – AUTHOR RESPONSE

Reviewer: 1

Reviewer Name: Gerda de Kuijper, MD PhD

Institution and Country: Centre for intellectual disability and mental health/Mental health care Drenthe (CVBP/GGZ Drenthe) the Netherlands

Please state any competing interests: None declared

Please leave your comments for the authors below

This study contributes importantly to the knowledge concerning the occurrence of movement side-effects caused by antipsychotic drug use in people with ID, in particular to the knowledge regarding the assumed greater sensitivity of people with ID compared to people without ID. Data source is valid and by using stratified sampling with frequency matching the sample will be representative for people with and without ID who use antipsychotics.

The results are presented clearly and are summarized in the tables and figure adequately.

We thank the reviewer for these positive comments.

I have some questions and remarks.

- In the Method section the dictionary of Read codes of the THIN database is mentioned. Since clinicians and researchers abroad may not be informed by the meaning of Read codes the authors are requested to explain this term.

Thank you for highlighting this. We have amended this paragraph to include more information about Read codes. A reference is included for those who require further detail.

- Method section Page 8. line 11-14: People also exited at the date they developed a movement side-effect as they were no longer considered at risk after this time.

Did the authors consider the possibility of the development of more/other movement side-effects after removal of the participant? How were incidence rates defined? The occurrence of one or more movement side-effects?

We have amended the manuscript to clarify our methods (now added in the 'statistical analysis' section).

- Incidence rate was defined as the number of new events (of recorded movement side-effect) / the duration that the cohort was at risk.

- First we were interested in the incidence of new cases of any movement disorder. Participants exited the cohort when they were first diagnosed with any movement disorder as they were no longer considered at risk of a new diagnosis after this date.

- For calculating the incidence of subtypes of movement side-effect, participants exited the cohort

after they were diagnosed with the type of movement side-effect of interest as they were no longer considered at risk of that type of movement side-effect after that date. They remained in the cohort for the purposes of being diagnosed with other types of movement side-effect as we understand that a single participant can develop more than one type of movement side-effect.

Page 8, line 39-42. In the Netherlands the anticholinergic agent biperiden is widely used to decrease symptoms of drug-induced Parkinsonism. Do UK physicians also prescribe this agent in this indication?

Biperiden is not listed in the British National Formulary and as such would not be available for General Practitioners to prescribe.

6. Are the outcomes clearly defined?

- In the Introduction section and in the Discussion section the word "Parkinsonism" is used to indicate the antipsychotic-induced Parkinson-disorder-like symptoms, while in the Method section the word "Pseudo-parkinsonism" is used. I would like to suggest to use the broadly accepted word "Parkinsonism" for antipsychotic-induced movement side-effects instead of the word "Pseudo-parkinsonism".

Thank you for highlighting this inconsistency. We have changed the manuscript and now use "Parkinsonism" throughout.

Furthermore, I would like to receive the Read code list for movement side-effects.

We are happy to now include the Read code lists for all movement side-effects as supplementary data.

Discussion section :

Page 13 , line 11-23

The rates of recording of tardive dyskinesia were relatively low....

The authors do not distinguish with regard to the level of severity of ID in the study sample. Studies have shown that more severe ID is associated with higher frequency and more severe symptoms of tardive dyskinesia. The mean severity of ID in this community dwelling study sample may be lower compared to other study samples of institutionalized participants.

Our previous work with THIN has demonstrated that degree of intellectual disability is unfortunately not reliably recorded in the database. We are therefore unable to use this as a variable in any analysis.

In the UK the vast majority of people with known intellectual disability live in the community (as opposed to institutions) and receive care from their General Practitioner. It is not likely, therefore, that the IQ of those people with intellectual disability in THIN is skewed towards the more mild range.

We have added to the 'data source' section of the paper to make this point clear to an international audience.

line 18-20...drug-induced tardive dyskinesia may be interpreted as part of the underlying ID and underrecorded, ... However, it may also be possible that the dyskinesia caused by the underlying ID is interpreted as drug-induced movement side-effect.

This is a valid point and we have added to the manuscript to reflect this possibility. This complexity reinforces the need to accurately monitor potential medication side-effects with thorough physical examination and regular review, ideally supported by validated side-effect scales.

12. Are the study limitations discussed adequately?

Page 14 line 37

In the manuscript is written that people with ID who use psychotropic drugs must consult their General Practitioner in case they suffer from drug-induced side-effects and that they may be less likely to present with side-effects compared to not ID people. Furthermore, the authors write that side-effects may have been missed if people with ID contact their psychiatrist directly. I wonder whether people with ID who use antipsychotics are being monitored for side-effects on a regular base in UK general practice and UK specialist practice. If not, the autors may provide some data on the number of people with ID who use antipsychotic drugs and consulted their physician for side-effects compared to those who use antipsychotics and did not consult their physician for side-effects.

We have discussed in the manuscript that it may be more or less likely that adverse side-effects of antipsychotic drugs are recognised and recorded in people with intellectual disability. Whilst on one hand people with intellectual disability may lack knowledge of side-effects and experience barriers to accessing care (such as complex booking systems or physical barriers), it is also possible that people with intellectual disability are more closely monitored as they are recognised as a group with a higher health need and receive additional medical input.

Good practice guidelines recommend that potential adverse side-effects of psychotropic medication are regularly monitored (e.g. Psychosis and schizophrenia in adults: prevention and management, NICE Clinical Guideline 178, February 2014). Real-life practice in this area is likely to vary and indeed Paton et al (BMJ Open 2016;6:e013116) demonstrated that fewer than 1 in 10 people with intellectual disability who are prescribed antipsychotic medication and who are under the care of secondary care services received a formal assessment for movement side-effects in the preceding year. We have added this reference to our manuscript and we end with a strong practice statement highlighting the need for regular and effective medication review in this group, including assessment of medication side-effects.

We are not aware of any national data on the rate of consultation for medication side-effects in people with intellectual disability that could be used as the reviewer suggests. Measuring this accurately using THIN is timely and beyond the scope of this study.

Reviewer: 2

Reviewer Name: Tamara Pringsheim

Institution and Country: University of Calgary, Canada

Please state any competing interests: None declared

Please leave your comments for the authors below

This is a very clearly written paper about the incidence of extrapyramidal symptoms in individuals taking antipsychotic medications using the THIN database, with the focus on comparing rates in individuals with intellectual disability versus those without intellectual disability.

This paper draws attention to the common, but now largely ignored issue of movement disorders in individuals exposed to antipsychotic medications. This paper demonstrates that not only are movement disorders more common in individuals with intellectual disability, but occur at a high frequency in all antipsychotic users, with 24% of the non-ID cohort having a history of movement

disorder and antipsychotic use at cohort entry. They have also shown that the incidence of recording of movement side effects in those prescribed antipsychotics fell over the course of the 15 year study period in both groups. I believe this is a generational effect, with an entire cohort of psychiatrists and physicians trained in the second generation antipsychotic era who have poor recognition of movement disorders.

I have no major concerns. I think this is a very important manuscript that deserves publication.

We wish to thank the reviewer for these supportive comments.

VERSION 2 – REVIEW

REVIEWER	Gerda de Kuijper, MD PhD Centre for Intellectual Disability and Mental Health/GGZ Drenthe, the Netherlands
REVIEW RETURNED	05-Jun-2017

GENERAL COMMENTS	I think the authors have addressed my questions and remarks sufficiently. They have amended and clarified a number of paragraphs, which makes the contents of the manuscript easier to understand. In my opinion the revised manuscript is suitable for publication in the BMP Open.
---